# Regulatory Role of Ribonucleotide Reductase Subunit M2 in Hepatocyte Growth and Pathogenesis of Hepatitis C Virus

**DOI:** 10.3390/ijms24032619

**Published:** 2023-01-30

**Authors:** Bouchra Kitab, Kyoko Tsukiyama-Kohara

**Affiliations:** Transboundary Animal Disease Center, Joint Faculty of Veterinary Medicine, Kagoshima University, Kagoshima 890-0065, Japan

**Keywords:** hepatitis C virus, RRM2, cell cycle, quiescence, viral oncogenesis

## Abstract

Hepatitis C virus (HCV) frequently causes chronic infection in the human liver, which may progress to advanced hepatic fibrosis, cirrhosis, and hepatocellular carcinoma. HCV primarily infects highly differentiated quiescent hepatocytes and can modulate cell cycle-regulatory genes and proliferation pathways, which ultimately contribute to persistent infection and pathogenesis. On the other hand, several studies have shown differential regulation of HCV RNA and viral protein expression levels, depending on the proliferation state of hepatocytes and the phase of the cell cycle. HCV typically requires factors provided by host cells for efficient and persistent viral replication. Previously, we found that HCV infection upregulates the expression of ribonucleotide reductase subunit M2 (RRM2) in quiescent hepatocytes. RRM2 is a rate-limiting protein that catalyzes de novo synthesis of deoxyribonucleotide triphosphates, and its expression is highly regulated during various phases of the cell cycle. RRM2 functions as a pro-viral factor essential for HCV RNA synthesis, but its functional role in HCV-induced liver diseases remains unknown. Here, we present a comprehensive review of the role of the hepatocyte cell cycle, in correlation with RRM2 expression, in the regulation of HCV replication. We also discuss the potential relevance of this protein in the pathogenesis of HCV, particularly in the development of hepatocellular carcinoma.

## 1. Introduction

The strict control of the expression levels of cell cycle regulatory proteins is essential for maintaining genomic stability and cellular homeostasis. In mammalian cells, ribonucleotide reductase subunit M2 (RRM2) is the rate-limiting protein in de novo synthesis of deoxyribonucleotide triphosphates (dNTPs) during DNA replication and repair, and therefore, plays a critical role in cellular proliferation [1]. RRM2 is ubiquitously expressed in human tissues, and its expression is strictly regulated throughout the cell cycle to preserve a balanced supply of dNTPs in the cell [1,2]. Because of this fundamental function, alterations in RRM2 expression have been linked to several human diseases, including DNA replication-enhanced diseases such as cancers [3].

Hepatitis C virus (HCV) is a positive-strand RNA virus that frequently establishes persistent infection in human hepatocytes, leading to chronic hepatitis and an increased risk of developing liver fibrosis, cirrhosis, and hepatocellular carcinoma (HCC). An estimated 56.8 million people worldwide have chronic HCV infection [4]. According to the latest global cancer statistics, HCC is the most frequent form of primary liver cancer, and HCV infection accounts for 20% of liver cancer-related deaths globally [5]. The development of highly effective direct-acting antivirals with high cure rates has been a major breakthrough in the management of HCV-related liver diseases. However, viral cure does not eliminate the risk of developing HCC, especially in patients with advanced liver fibrosis and cirrhosis [6,7]. This suggests that HCV-induced mechanisms that drive hepatocarcinogenesis persist even after virus elimination. The molecular mechanisms underlying HCV-induced liver diseases and HCC development, especially after viral cure, are not completely known [8]. HCV has a cytoplasmic lifecycle. The 9.6 kb viral RNA genome contains one open reading frame, which translates into a single polyprotein [9]. The translation is mediated by a highly conserved internal ribosome entry site (IRES) in the 5’-untranslated region of the viral RNA [10]. Post-translational cleavage of the polyprotein by cellularly and virally encoded proteases yields 10 structural and nonstructural (NS) proteins, in the following order: NH_2_-core-envelope(E1)-E2-p7-NS2-NS3-NS4A-NS4B-NS5A-NS5B-COOH [11]. Numerous studies have shown that HCV infection and viral proteins may modulate cell cycle-controlling pathways, by affecting cellular functions or the levels of cell cycle regulatory proteins, which can have detrimental effects on host cell physiology and contribute to HCV-induced liver diseases [12,13,14]. Previously, we showed that *RRM2* expression was upregulated in response to HCV infection in quiescent hepatocytes [15]. RRM2 may function as a pro-viral factor essential for HCV RNA synthesis, by enhancing the stability of the HCV RNA-dependent RNA polymerase NS5B [15]. It remains unclear whether RRM2 has functional implications in hepatitis C pathogenesis and HCC development. Nevertheless, there is growing evidence that RRM2 is overexpressed in HCC patients, and inhibition of *RRM2* expression using small interfering RNA (siRNA) significantly reduces HCC cell proliferation in vitro [16,17,18]. Although there is still limited knowledge, continuous research in this area may ultimately lead to the identification of effective therapeutic options, especially against HCC.

## 2. RRM2, Its Expression throughout the Cell Cycle, Subcellular Localization, and Functions

RRM2 serves as the regulatory subunit of the ribonucleotide reductase (RNR) holoenzyme, which catalyzes the rate-limiting step in de novo dNTPs synthesis, by reducing ribonucleoside diphosphates to 2′-deoxy ribonucleoside diphosphates, the phosphorylation of which yields dNTPs [1]. In mammalian cells, RNR exists as a heterodimeric tetramer composed of two identical large catalytic subunits (ribonucleotide reductase M1 [RRM1]) and two identical small regulatory subunits (RRM2) [1]. Each RRM1 subunit contains two allosteric sites and a catalytic site which is formed only in the presence of the RRM2 subunit [19]. Each RRM2 subunit contains a non-heme iron center and tyrosyl-free radical [20,21]. Both RRM1 and RRM2 are essential for catalysis [1].

RNR activity is tightly regulated during the cell cycle, to preserve the balanced supply of dNTPs in the cell, which is essential for maintaining cell survival and genome stability. This regulation is mainly dependent on RRM2 protein expression [22]. Indeed, the transcription of *RRM1* and *RRM2* genes in proliferating cells is induced at the G1-to-S-phase transition, reaching the highest level during the S-phase, when dNTPs are needed for nuclear DNA replication and repair [23]. Studies have shown that the *RRM2* promoter is activated by the transcription factors NF-Y [24] and E2F [25]. It has also been reported that the gene transcription of *RRM1* and *RRM2* is regulated by cyclin-dependent kinases (CDKs) during the G1- and S-phases [1,26]. While RRM1 protein level remains relatively constant and in excess throughout all phases of the cell cycle, because of its long half-life of 15 h, RRM2 protein has a short half-life of 3 h, and its level oscillates during the cell cycle: low or undetectable in G0/G1-phase, rising in late G1-phase, peaking in S-phase, and finally declining in G2- and M-phases [27,28]. The low levels of RRM2 protein outside the S-phase have been attributed to its proteasome-mediated degradation by the anaphase-promoting complex/Cdh1 ubiquitin ligase, during the G0/G1-phase and late mitosis [29], and Skp1–Cullin–F-box (SCF)^Cyclin F^ ubiquitin ligase, during the G2-phase [26]. Thus, the RRM2 subunit is the primary regulator of RNR activity in proliferating cells. In quiescent or non-dividing cells, the *RRM2* gene is neither transcribed nor upregulated after DNA damage [28]. An additional small subunit RRM2B (or p53R2), which is induced by the tumor suppressor protein p53, substitutes RRM2 to form a highly active RNR complex [30,31]. p53R2, together with RRM1, provides dNTPs for nuclear DNA repair and mitochondrial DNA replication in quiescent cells [32].

Regulation of the subcellular localization of RRM2 in mammalian cells provides an additional level of control of RNR activity. Early immunochemical studies have demonstrated exclusive cytoplasmic localization of RRM2 and RRM1 in S-phase cells [2,33]. In line with these data, Pontarin et al. showed that ribonucleotide reduction in the cytoplasm is mediated by both the RRM1/RRM2 and RRM1/p53R2 complexes, and suggested that dNTPs produced by the enzyme diffuse into the nucleus or are transported into the mitochondria for DNA synthesis [34]. However, there is increasing evidence that RRM2 translocates from the cytoplasm to the nucleus in response to DNA damage, to ensure local availability of dNTPs at DNA damage sites, for efficient DNA repair [26,28,35]. Furthermore, D’Angiolella et al. reported that during the G2 phase, RRM2 enters the nucleus to interact with cyclin F and is degraded [26].

## 3. Cell Cycle-Dependent Regulation of HCV RNA Replication

Under normal physiological conditions, most hepatocytes in the adult human liver are highly differentiated and maintained in a quiescent or non-dividing state, known as the G0-phase of the cell cycle [36]. During natural infection, HCV infects and replicates in quiescent human hepatocytes, resulting in persistent and productive infection [37,38]. However, attempts to establish the relationship between HCV replication and hepatocyte proliferation state have shown a tight coupling of HCV viral RNA and protein levels to host cell growth. Using subgenomic or full-length HCV replicon cells, several studies have demonstrated that the highest levels of HCV RNA and viral protein production were detected in actively dividing and sub-confluent human hepatoma HuH-7 cells, whereas a sharp decrease in HCV RNA and protein levels was observed when replicon cells reached confluence [39,40,41]. In support of these results, other studies have reported that HCV viral RNA synthesis decreased strongly in poorly proliferating, confluent, or serum-starved cells, and was enhanced in the S-phase of the cell cycle [42,43]. Growth arrest of confluent cells has been proposed to be responsible for the confluence-based inhibition of HCV replication [41]. Furthermore, one study reported that reduced intracellular pools of nucleotides account for the decrease in HCV replication in confluent cells, possibly because these cells depend on salvage nucleoside biosynthesis pathways, and thus, have a lower level of intracellular nucleotide triphosphates [44].

These observations of HCV dependency on cell density or proliferation state indicate the presence of host factors that vary in abundance during the hepatocyte cell cycle and play a critical role in HCV replication. Evidence for the role of such factors comes from the previous findings of Honda et al., which showed that HCV core protein-expressing cells entered the S-phase earlier than non-core-expressing cells, indicating that HCV can induce quiescent cells to enter the cell cycle, possibly creating an environment that generates host factors required for efficient viral replication [45]. The same research group also reported that IRES-mediated HCV RNA genome translation was greatest in the M-phase of the cell cycle and lowest in the quiescent G0-phase of resting cells [42]. Nevertheless, Nelson and Tang reported that cell growth arrest does not automatically lead to inhibition of HCV replication, which supports the ability of HCV to replicate in the human liver, where most hepatocytes are quiescent [41]. Subsequently, the inflammatory microenvironment established in the human liver, to clear HCV-infected cells, leads to repeated cycles of hepatic injury and regeneration, resulting in an increased number of actively dividing cells, higher IRES activity, and enhanced HCV RNA replication [42]. It is evident that hepatocyte changes that occur upon entry into the cell cycle are likely to be advantageous for persistent HCV infection, with more abundance of the host cell cycle machinery and factors required for viral RNA replication [13,42].

## 4. The Link between RRM2 Expression, S-Phase of Cell Cycle, and HCV RNA Synthesis

As stated above, hepatocytes are mostly in a non-dividing state in vivo, and thus, do not express *RRM2*. Our previous study used genome-wide microarray analysis and showed that *RRM2* expression is upregulated in response to HCV infection in quiescent hepatocytes from chimeric mice with humanized livers [15]. Upregulation of RRM2 mRNA and protein levels was also confirmed in hepatoma cells infected with the HCV strain JFH1. RRM2 has been shown to be required for HCV RNA replication, as siRNA-mediated silencing of *RRM2* suppressed HCV replication and infection, while *RRM2* overexpression rescued the same [15]. Scholle et al. previously reported that HCV viral RNA synthesis is highly stimulated and enhanced in the S-phase of the cell cycle [43]. Thus, it is possible that elevated RRM2 expression may reflect the increased number of hepatocytes that enter the S-phase in response to HCV infection. Further investigations are required to clarify this hypothesis. Furthermore, we found that RRM2 directly interacts with the HCV RNA-dependent RNA polymerase, NS5B. RRM2 unpairs the ubiquitin-mediated proteasomal degradation of NS5B, resulting in enhanced stability of the NS5B protein, thereby promoting viral RNA synthesis [15] (Figure 1).

Munakata et al. demonstrated that NS5B forms a complex with the retinoblastoma tumor suppressor protein (pRb), thereby targeting it for ubiquitination and proteasome-dependent degradation [46,47]. pRb is a master regulator of the cell cycle and plays a major role in controlling the G1- to S-phase transition and mitotic checkpoints, by repressing E2F transcription factors [48]. Decreased pRb protein levels in HCV-infected cells result in the activation of E2F-responsive promoters and transcriptional activation of genes required for S-phase and DNA replication, which stimulates entry into the S-phase and cellular proliferation [46] (Figure 1). Meanwhile, the *RRM2* gene contains an E2F-binding site and can be activated at the transcriptional level by E2F1 overexpression in quiescent cells, before the induction of the S-phase [25]. Our group previously demonstrated that cells expressing full-length HCV RNA activate the CDK-pRb-E2F pathway [49]. These data suggest that *RRM2* transcription is mainly dependent on the E2F transcription factor during HCV infection. Moreover, it has been shown that NS5B delays cells in the S-phase, through interaction with cyclin-dependent kinase 2-interacting protein (CINP) and relocalization of the protein from the nucleus to the cytoplasm [50]. CINP relocalization affects signaling pathway molecules involved in cell cycle regulation, as shown by the reduction of pRb and phosphorylated checkpoint kinase 1 (pChk1) [50]. Chk1 is an evolutionarily conserved serine/threonine kinase that is essential for genomic maintenance and serves as a cell cycle checkpoint protein during the S- and G2/M-phases [51,52]. Chk1 is phosphorylated by ataxia telangiectasia and RAD3-related (ATR) protein kinase, an S-phase-specific sensor/transducer that is activated in response to ultraviolet light, hydroxyurea, and replication stress, following genotoxic damage [53]. Several previous studies have demonstrated that downregulation or inactivation of Chk1 leads to S-phase arrest [54,55]. Naruyama et al. observed S-phase arrest after Chk1 depletion, caused by transcriptional suppression of *RRM2* [56] (Figure 1). These alterations ultimately result in an S-phase delay via the DNA damage/repair pathway.

**Figure 1 ijms-24-02619-f001:**
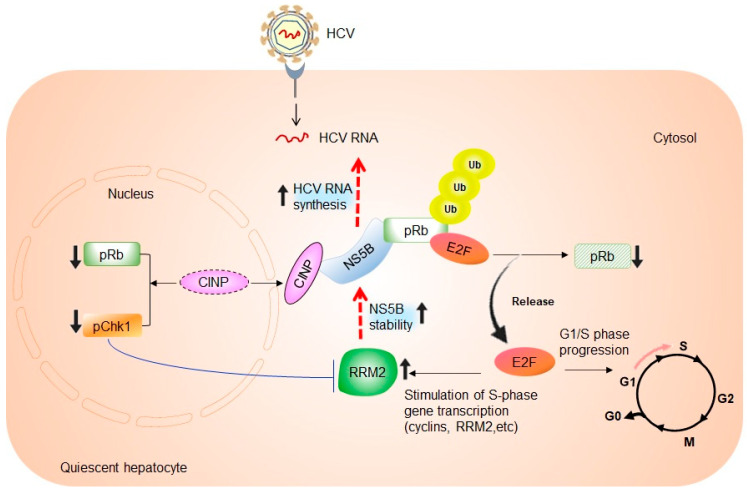
Role of the RRM2-NS5B interaction in regulating hepatocyte cell cycle regulation and proliferation. NS5B protein forms a complex with the retinoblastoma tumor suppressor protein (pRb), thereby targeting it for ubiquitination and proteasome-mediated degradation. This results in the release of the E2F transcription factor and activation of E2F-responsive promoters. Likewise, there is a stimulation of transcription of S-phase genes, including RRM2, G1- to S-phase progression, and therefore, cell proliferation [46]. Increased RRM2 expression enhances the stability of the NS5B protein, and in turn, HCV RNA replication [15]. On the other hand, NS5B interacts with the cyclin-dependent kinase 2-interacting protein (CINP), resulting in its relocalization from the nucleus to the cytoplasm. CINP relocalization results in the reduction of phosphorylated pRb and checkpoint kinase 1 (pChk1) abundance levels [50]. Chk1 depletion induces transcriptional suppression of *RRM2*. This may result in S-phase delay and suppression of cell proliferation [56].

In addition to NS5B, other HCV proteins have been shown to modulate the entry of hepatocytes into the S-phase. One study showed that HCV core protein expression increased the fraction of human hepatoblastoma HepG2 cells in the S-phase, by increasing the stability of the c-myc oncoprotein [57]. Another study demonstrated that NS2 can inhibit cell proliferation and induce cell cycle arrest in the S-phase. The induction of S-phase arrest in NS2-expressing cells is associated with decreased cyclin A expression [58]. In murine fibroblasts and HepG2 cells, NS5A protein repressed the transcription of the cyclin-dependent kinase inhibitor p21 and increased the expression of proliferating cell nuclear antigen, which is expressed during the S-phase and required for DNA replication [59]. Overall, the results of these studies demonstrate that HCV proteins can modulate the levels or functions of cell cycle-regulatory proteins, and such effects might ultimately contribute to HCV persistence and pathogenesis.

## 5. Potential Relevance of HCV-Induced RRM2 Upregulation in Promoting Liver Tumorigenesis

The relationship between RRM2 expression and tumorigenesis is well-established and remains the subject of intense research [3,60,61]. Aye et al. surveyed RNR gene expression in human cancers using the ONCOMINE cancer microarray database and found that RRM2 was among the top 10% most overexpressed genes in 73 out of the 168 cancer analyses [3]. These include sarcoma and cancers of the bladder, brain, central nervous system, breast, colorectal, liver, and lung [3]. Elevated RRM2 levels have been correlated with cellular invasiveness, metastasis, tumorigenesis, and poor prognosis [62,63,64]. Furthermore, RRM2 has been identified as a tumor promoter and cancer therapeutic target. Carcinogenesis due to RRM2 overexpression is not only related to enhanced genomic instability and mutagenesis, due to imbalanced production of dNTPs, but also the ability of RRM2 to induce activation of several oncogenes, including v-fms, v-src, A-raf, v-fes, c-myc, and NF-kB, which promote cell transformation and tumorigenesis [26,63,65]. There is also evidence that RRM2 overexpression promotes angiogenesis, whereas its downregulation induces apoptosis and G1-phase arrest, in addition to inhibiting cell proliferation in various cancer types [60,66,67].

RRM2 is believed to play a vital role in the development of HCC. Immunohistochemical staining of RRM2 protein in HCC and normal liver tissues showed that 2 out of 10 HCC cases (20%) had high/medium RRM2 staining, which was the highest among 20 other cancer types. In contrast, RRM2 protein expression was undetectable in normal liver tissues [18]. Other in vitro experiments have shown that RRM2 is overexpressed in all human HCC cell lines, including SMMC-7721, HepG2, HuH-7, HCC-LM3, HCC-97L, Hep3B, and PLC/PRF/5 [68]. Using immunohistochemistry in tumor tissues from HCC patients who underwent curative hepatectomy, with viral and non-viral causes of HCC, one study showed the high RRM2 expression in 19 of 25 (76%) HCV-related HCC patients correlated with their poor prognosis [16]. Several groups have used public databases and bioinformatics methods to identify differentially expressed genes related to the progression, diagnosis, and prognosis of HCC. Zhou et al. compared microarray expression profiles between HCV-related HCC and their matched non-cancerous tissues and then selected genes with high values based on the genome-wide relative significance and genome-wide global significance models [69]. Two clusters, 3 and 4, with higher GWGS values were identified. In cluster 3, *RRM2* was among the top nine genes reported to play a key role in HCC [69]. The risk for HCC is highest in HCV patients with cirrhosis. He et al. compared microarray data of HCC and cirrhosis tissues from patients with hepatitis B virus infection, HCV infection, or hepatitis B virus and hepatitis D virus co-infection, to identify pivotal genes and pathways involved in HCC transformed from cirrhosis [70]. They found that *RRM2*, cyclin-dependent kinase 1 (*CDK1*), PDZ-binding kinase (*PBK*), abnormal spindle homolog, and microcephaly-associated Drosophila (*ASPM*) were key genes for HCC transformation from cirrhosis. In addition, two significantly enriched pathways were identified: cell cycle and p53 signaling pathways. CDK1 is enriched in the cell cycle and p53 signaling pathways, while RRM2 is enriched in the p53 signaling pathway [70]. This suggests that CDK1 affects HCC via its roles in the cell cycle and p53 signaling pathways, whereas RRM2 affects HCC via its role in the p53 signaling pathway. The main results of the studies that have investigated RRM2 expression in HCV-related HCC are summarized in Table 1.

The exact role of RRM2 in HCV-associated hepatocarcinogenesis has not yet been defined; however, different mechanisms can be proposed, which require further experimental validation (Figure 2). RRM2 contains an oxygen-bridged dinuclear iron center that generates a tyrosyl-free radical essential for RNR activity [1]. High RRM2 levels result in increased free radical production and reactive oxygen species (ROS) formation, which causes oxidative stress and DNA damage, leading to cell transformation [3]. One study showed that RRM2 induces ROS production to activate the extracellular signal-regulated kinase 1/2 pathway, which promotes cervical cancer angiogenesis related to human papillomavirus infection [71]. ROS plays a critical role in HCV-related hepatocarcinogenesis, and the HCV viral proteins core, NS3, and NS5A are potent inducers of oxidative stress [72,73,74]. In HCV core-transgenic mice, the core protein induces HCC by altering the oxidant/antioxidant state [75]. Furthermore, HCV can impair the function of the tumor suppressor protein p53 by inducing 3β-hydroxysterol∆24-reductase (DHCR24) overexpression, thereby suppressing stress-induced apoptosis and promoting tumorigenicity [76]. Hence, it is possible that RRM2 upregulation represents another pathway for HCV-induced oxidative stress. In addition, as outlined above, RRM2 enhances the stability of HCV NS5B polymerase, which promotes degradation of the tumor suppressor protein pRb, resulting in the activation of E2F-responsive promoters, and therefore, cell proliferation [15,46]. pRb is frequently inactivated in HCC, and HCV-mediated regulation of pRb levels is one of the primary mechanisms involved in the high incidence of HCC in HCV-infected patients [49]. Activation of the phosphatidylinositol-4,5-biphosphate 3-kinase (PI3K)/AKT/mechanistic target of rapamycin (mTOR) signaling pathway has been described in many malignancies, including HCC, with its role in promoting persistent HCV infection also reported [77]. The AKT/mTOR signaling pathway positively controls both the gene transcription and protein translation of RRM2, whereas the same is suppressed by p53 signaling [78]. Importantly, *RRM2* silencing leads to the inactivation of the PI3K-Akt-mTOR pathway, resulting in S-phase arrest and apoptosis [79]. These findings suggest reciprocal regulation of the RRM2 and PI3K/AKT/mTOR pathways in tumorigenesis. Interestingly, abundant profiles have indicated that microRNAs play important roles in HCC proliferation, progression, and metastasis, by regulating key signaling pathways involved in cellular proliferation, apoptosis, invasion, and angiogenesis [80,81]. Numerous microRNA profiles in HCC have been reported [80]. A recent study showed an interplay between miR-582-3p and RRM2 in the regulation of HCC progression [82]. RRM2 promotes oncogenic effects in HCC by triggering Wnt/β-catenin signaling, whereas miR-582-3p depletes RRM2 expression, thereby impairing the activation of Wnt/β-catenin signaling and blocking the progression of HCC [82].

## 6. Developing RRM2 as Molecular Biomarker and Therapeutic Target in HCV-Associated HCC

The primary prevention strategy for HCC is to eliminate HCV infection using antiviral therapy. The development of a highly effective direct-acting antiviral-based combination therapy has offered incredible success in the treatment of HCV infections. However, several studies have claimed that the risk of HCC is decreased, but not eliminated, after viral cure, especially in patients with advanced hepatic fibrosis [6,7]. Among direct-acting antivirals, sofosbuvir constitutes the mainstay of most anti-HCV combination regimens [83]. Sofosbuvir is a uridine analog that specifically targets HCV NS5B polymerase and inhibits viral RNA synthesis [84]. Similar to sofosbuvir, gemcitabine is a nucleoside analog and has been reported to inhibit HCV infection [85]. Gemcitabine was shown to inactivate ribonucleotide reductase activity by interfering with RRM2 and binding to RRM1 by its diphosphate form, and therefore, has been approved for the treatment of various types of cancer including HCC [86]. From these data, a possible interference between direct-acting antivirals and RRM2 can be generated and would be an interesting issue for further investigations.

HCC has a poor prognosis and low survival rate owing to its delayed diagnosis, drug resistance, and frequent recurrence [87]. Accordingly, there is a need for the development of convenient biomarkers to predict HCC risk after viral cure, in addition to strategies for HCC prevention. Lee et al. elucidated the prognostic significance of RRM2 protein expression in a large cohort of HCC patients with long-term follow-up. They found that high RRM2 expression was significantly associated with early recurrence and intrahepatic metastasis in HCC patients and suggested that high RRM2 expression could be a useful biomarker to predict early recurrence of HCC [16]. Another analysis of *RRM2* mRNA levels in HCC patients and their impact on the overall survival of patients showed that higher *RRM2* mRNA expression was significantly correlated with lower overall survival [18].

There is growing evidence that targeting RRM2 and its downstream signaling pathways is a rational approach for developing novel anti-HCC- and anti-HCV-related HCC therapeutics. Through a combination of genome-wide expression and functional screening, Satow et al. selected RRM2 as a promising candidate therapeutic target for HCC [17]. Furthermore, siRNA-mediated knockdown of *RRM2* inhibited the proliferation of HCC cells and the growth of HCC xenografts transplanted into immunodeficient mice [17]. siRNA duplexes are another class of nucleic acids that are capable of achieving potent sequence-specific inhibition of gene expression. Heidel et al. developed a potent siRNA duplex against *RRM2*, siR2B + 5, and reduced the growth potential of cancer cells, both in vitro and in vivo [60]. The role of RRM2 inhibitors in the treatment of HCC has been less studied. One study showed that the approved choleretic drug, osalmid, exhibits much higher RRM2-inhibitory activity than the RRM2 inhibitor hydroxyurea, and significantly suppresses HBV replication in vivo and in vitro [88]. Osalmid inhibits RRM2 activity by competitively binding to its RRM2 active site. Another metabolite of osalmid (M7) significantly suppresses human HCC progression by inhibiting RRM2 activity and inducing cell cycle arrest and apoptosis via p53-related signaling pathways [89]. The importance of multi-kinase and immune checkpoint inhibitors in HCC treatment has been clearly demonstrated. Sorafenib, an oral multi-kinase inhibitor, was the first molecular-targeted therapy approved for advanced HCC in the first-line setting [90]. Sorafenib exerts both anti-angiogenic and apoptosis-inducing effects in HCC [91]. Recently, RRM2 was identified as a target of sorafenib, and inhibition of RRM2 mRNA and protein expression is a common effect of sorafenib, which partially contributes to its anticancer activity in HCC cells [18]. Further experiments revealed that sorafenib downregulates RRM2 expression by suppressing its transcription and promoting its protein degradation [18]. In addition, inhibition of *RRM2* expression using siRNA significantly reduced HCC cell proliferation, which was comparable to the anticancer activity of sorafenib [18]. The downregulation of RRM2 by sorafenib may provide additional clinical benefits. Sorafenib has been shown to induce ferroptosis, which significantly prolongs the survival of advanced HCC patients [92]. Ferroptosis is a type of regulated cell death, suggesting that ferroptosis induction may provide a promising approach for the efficient triggering of cancer cell death. RRM2 has been identified as an endogenous ferroptosis suppressor [18]. A recent study showed that inhibiting fatty acid desaturase 2 (FADS2), a key determinant of cellular sensitivity to ferroptosis, markedly enhances HCV replication, whereas the ferroptosis-inducing compound erastin sensitizes HCV to direct-acting antiviral agents [93]. It is possible that the RRM2-inhibitory activity of sorafenib results in the induction of ferroptosis, which provides a promising approach for simultaneously triggering HCC cell death and inhibiting HCV replication.

## 7. Conclusions

HCV has evolved different strategies to ensure cellular conditions that are beneficial for viral replication and long-term survival in the host cell. RRM2 is a critically important protein for DNA synthesis and cell cycle control, and its expression is exclusively stimulated during the S-phase in cells undergoing the cell cycle. In response to HCV infection, RRM2 expression, which promotes viral RNA synthesis, is upregulated in quiescent hepatocytes. Since there is evidence indicating a strong relationship between HCV infection and the cell cycle status of hepatocytes, it can be speculated that RRM2 is greatly involved in this relationship. In contrast, dysregulation of RRM2 expression may influence various signaling pathways involved in cell growth, survival, and migration, which play a crucial role in tumorigenesis. Although there are no specific studies on the direct implication of RRM2 in HCV-related HCC, the available data suggest that RRM2 may play an essential role in the pathogenesis of persistent HCV infection. There is a need for further investigations to clarify the mechanisms and functions of RRM2 in HCV-related liver diseases, particularly in HCC.

## Figures and Tables

**Figure 2 ijms-24-02619-f002:**
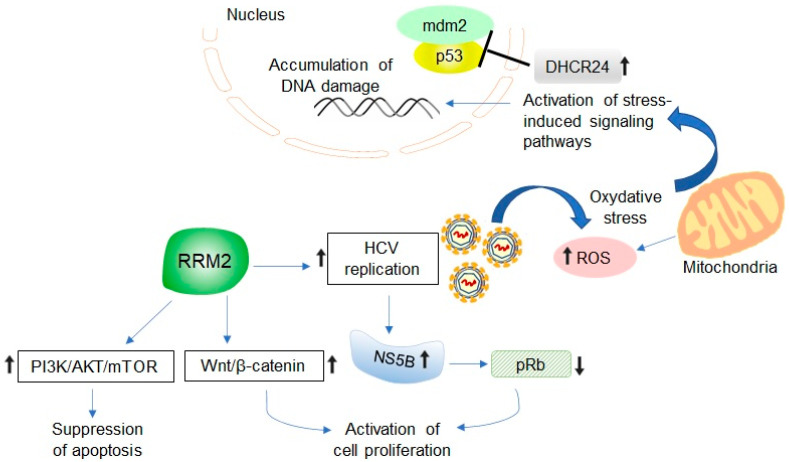
Schematic model of possible viral mechanisms and host signaling pathways induced by upregulated RRM2 expression and may contribute to HCV-associated liver tumorigenesis.

**Table 1 ijms-24-02619-t001:** Main studies that have investigated RRM2 expression in HCV-associated HCC.

First Author [Ref.]	Samples and Methodology	Main Results
Lee et al. [16]	RRM2 protein expression was evaluated using immunohistochemistry in tumor tissues from 259 HCC patients who underwent curative hepatectomy: - 198 HBV-HCC patients - 25 HCV-HCC patients - 36 patients with non-viral causes of HCC	High RRM2 expression was observed in 210 of 259 patients (81.1%): - 167/198 (84.3%) HBV-HCC patients - 19/25 (76%) HCV-HCC patients - 24/36 (66.7%) non-viral HCC patients High RRM2 expression was significantly associated with viral etiology and liver cirrhosis. High RRM2 expression was correlated with poor prognosis and early recurrence of HCC after hepatectomy.
Zhou et al. [69]	Three microarray expression profiles corresponding to HCV-related HCC and non-cancerous tissue samples were selected. Genes with high values were identified based on GWRS and GWGS models.	Two clusters, 3 and 4, had higher GWGS values. In cluster 3, nine genes were found to play a key role in HCC: *RRM2*, *PTTG1*, *CDK1*, *CDKN2A*, *CDKN3*, *ASPM*, *ADAMTS13*, *ECM1*, and *CXCL12*. Differentially expressed genes, including *RRM2*, *CDK1*, *PBK*, and *ASPM*, encoded the hub proteins with higher degrees in the PPI network.
He et al. [70]	Bioinformatics analysis of microarray data from 20 liver cirrhosis samples and 17 HCC samples. The etiology of these samples was HBV infection, HCV infection, or HBV + HDV co-infection.	*CDK1* and *RRM2* may be key genes related to the development of HCC from cirrhosis. Two enriched pathways were identified, including the cell cycle and p53 signaling pathways. RRM2 was enriched in the p53 signaling pathway.

GWGS, genome-wide global significance; GWRS, genome-wide relative significance; *RRM2*, ribonucleotide reductase M2; *PTTG1*, pituitary tumor-transforming gene 1; *CDK1*, cyclin-dependent kinase 1; *CDKN2A*, cyclin-dependent kinase inhibitor 2A; *CDKN3*, cyclin-dependent kinase inhibitor 3; *ASPM*, assembly factor for spindle microtubules; *ADAMTS13*, ADAM metallopeptidase with thrombospondin type 1 motif 13; *ECM1*, extracellular matrix protein 1; *CXCL12*, C-X-C motif chemokine ligand 12; PBK, PDZ-binding kinase; PPI, protein-protein interaction network; HBV, hepatitis B virus; HCV, hepatitis C virus; HDV, hepatitis D virus.

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
