# Peer review of "Regulatory Role of Ribonucleotide Reductase Subunit M2 in Hepatocyte Growth and Pathogenesis of Hepatitis C Virus"

_ijms, 2023, doi:10.3390/ijms24032619_

Round 1

Reviewer 1 Report

This is an review article about a ribonucleotide reductase holoenzyme RRM2 and its role in HCV associated HCC. The review was comprehensive and the writing is in a fashion style, which made the article easy to read. 

The reviewer suggests some more points of view to discuss in this review as following:

1. Currently, direct-acting antiviral (DAA) treatment of HCV is more than 99% effective. Whether one of the most used DAA, sofosbuvir, which is converted to a nucleotide monophosphate analog intracellularly, may interact with of be interfered by RRM2 could be an interesting point to be included in this review.

2. Could DNA damage perturb HCV replication due to the lack of dNTP?

3. Does HCV prefer S phase, at which cell cycle that IRES-dependent translation is more dominant than the cap-dependent ones?

Author Response

Thank you very much for your valuable comments. We have addressed their questions and responded, as followings.

Reviewer 1

  1. Currently,direct-acting antiviral (DAA) treatment of HCV is more than 99% effective. Whether one of the most used DAA, sofosbuvir, which is converted to a nucleotide monophosphate analog intracellularly, may interact with of be interfered by RRM2 could be an interesting point to be included in this review.

Thank you for your comment. We have discussed the possible interference between RRM2 and nucleotide analogs such as sofosbuvir in the section 6 of the manuscript (line 343-351).

  1. Could DNA damage perturb HCV replication due to the lack of dNTP?

Thank you for your comment. It has been shown that HCV replication induces DNA damage stress and activates DNA damage signaling pathways. The ataxia-telangiectasia mutated kinase (ATM) and the checkpoint kinase 2 (Chk2) are key regulators of the DNA damage response pathways. Activation of these kinases in response to damaged DNA results in the activation of cellular processes that integrates cell-cycle control and DNA repair or apoptosis. Ariumi et al (Journal of virology, 2008, p. 9639–9646) examined the level of HCV RNA in HuH-7 cells expressing short hairpin RNA targeted to ATM or Chk2. The authors showed that HCV replication is suppressed in ATM- or Chk2-knockdown cells. In addition, treatment with ATM kinase inhibitor suppressed HCV RNA replication(Ariumi et al, Journal of virology 2008). These data suggest that the level of DNA damage is critical for HCV RNA replication.

  1. Does HCV prefer S phase, at which cell cycle that IRES-dependent translation is more dominant than the cap-dependent ones?

Thank you very much for your comment. According to our knowledge, there are no studies clearly describing the preference of HCV to a specific cell-cycle phase for efficient and persistent infection in the host cell. As outlined in our manuscript, several studies investigated the relationship between HCV replication and hepatocyte proliferation state and therefore cell cycle. According to a study by Honda et al. (Gastroenterology 2000;118:152–162), it has been shown that compared with cap-dependent translation, the activity of HCV-IRES mediated translation was highest during the mitotic (M) phase. Other group reported that HCV-IRES dependent translation activity was highest during G0 and G1 phase but significantly drops during S phase (Cell Cycle 11:2, 277-285, 2012), which might be concerned with ectopic miR-122 works.

Reviewer 2 Report

Bouchra Kitab and Kyoko Tsukiyama-Kohara summarized the role of RRM2 in regulation of HCV replication and showed relevance of RRM2 in promoting liver tumorigenesis. The manuscript is well written, and I believed that the manuscript is interesting.  I have a minor comment listed below. In the section 5, the authors highlighted the correlation between RRM2 expression and tumorigenesis. RRM2 was shown to be involved in Wnt/β-catenin signaling and PI3K/AKT/mTOR pathways et al. that plays role in hepatocarcinogenesis. I was wondering if the authors could summarize all these information in a table or figure.

Author Response

Thank you very much for your valuable comments. We have addressed their questions and responded, as followings.

Reviewer 2

Bouchra Kitab and Kyoko Tsukiyama-Kohara summarized the role of RRM2 in regulation of HCV replication and showed relevance of RRM2 in promoting liver tumorigenesis. The manuscript is well written, and I believed that the manuscript is interesting.  I have a minor comment listed below. In the section 5, the authors highlighted the correlation between RRM2 expression and tumorigenesis. RRM2 was shown to be involved in Wnt/β-catenin signaling and PI3K/AKT/mTOR pathways et al. that plays role in hepatocarcinogenesis. I was wondering if the authors could summarize all these information in a table or figure.

Based on the reviewer’s instruction, we have summarized the various cellular and viral mechanisms related to the role RRM2 in HCV-induced liver carcinogenesis in Figure 2.  
